# Pre-Trauma Pain Is the Strongest Predictor of Persistent Enhanced Pain Patterns after Severe Trauma: Results of a Single-Centre Retrospective Study

**DOI:** 10.3390/medicina59071327

**Published:** 2023-07-19

**Authors:** Katharina Fetz, Rolf Lefering, Sigune Kaske

**Affiliations:** 1Institute for Research in Operative Medicine (IFOM), Witten/Herdecke University, 51109 Cologne, Germany; 2Chair of Research Methodology and Statistics, Department of Psychology, Witten/Herdecke University, 58448 Witten, Germany; 3Department of Anaesthesiology and Operative Intensive Care, Cologne Merheim Medical Centre, 51109 Cologne, Germany; 4Institute for Emergency Medicine, University Hospital Schleswig-Holstein, 24118 Kiel, Germany; 5Department of Trauma Surgery, Cologne Merheim Medical Centre, 51109 Cologne, Germany

**Keywords:** trauma, pain, chronic pain, persistent pain, health-related quality of life

## Abstract

*Background and Objectives:* Traumatic injuries are a significant public health issue worldwide, with persistent enhanced pain being a common complication following severe trauma. Persistent and chronic pain can have a profound impact on patients’ quality of life, affecting physical, emotional, and social functioning. This study aimed to investigate the pain patterns of trauma patients before and after severe trauma, and identify the predictors of persisting pain after injury. *Materials and Methods:* A total of 596 patients of a level-one trauma centre with severe trauma were included in this study. The Trauma Outcome Profile Scale was used to assess pain severity before and after trauma, and a logistic regression analysis was performed to determine the most significant predictors of relevant pain after severe trauma. *Results:* The mean age of the included patients was 48.2 years, and 72% were males. The most frequent cause of injury was traffic accidents, and the mean Injury Severity Score was 17.6. Nearly half of the patients experienced reduced pain-related quality of life after trauma, with persisting pain predominantly occurring in the neck, spine, shoulder, pelvis, hip, knee, and feet. Even minor injuries led to increased pain scores. Preexisting pain before injury (OR: 5.43; CI: 2.60–11.34), older age (OR: 2.09, CI: 1.22–3.27), female gender (OR: 1.08, CI: 0.73–1.59), and high injury severity (OR: 1.80, CI: 1.20–2.69) were identified as significant predictors of enhanced pain. *Conclusions*: These findings highlight the importance of considering pre-existing pain, body area, and injury severity in assessing the risk of persistent pain in trauma patients.

## 1. Introduction

Traumatic injuries pose a significant global public health challenge, encompassing a broad spectrum of injuries arising from accidents, violence, or self-infliction. Among individuals aged 5–44 years, injuries resulting from traffic accidents, falls, and violence are the primary cause of death, contributing to more than 10% of global fatalities [1]. Furthermore, traumatic injuries can lead to long-term disabilities, chronic pain, and diminished quality of life, profoundly impacting both individuals and their families [2,3]. The economic implications of traumatic injuries are substantial, encompassing substantial costs associated with medical care, rehabilitation, and productivity loss [4]. Given the widespread prevalence of traumatic injuries and their profound influence on public health, further research is necessary to enhance prevention, treatment, and rehabilitation strategies, thereby alleviating the burden these injuries impose on individuals, families, and society at large [5].

Pain following severe trauma can have a significant impact on the patient’s quality of life [6,7,8,9,10]. According to Keene et al. [10], up to two-thirds of major trauma victims report ongoing pain severe enough to affect their quality of life for several years after injury. Chronic pain is a common complication following polytrauma and can lead to physical, emotional, and social limitations. It can affect the ability to perform daily activities, increase dependency on others, and cause financial burden due to medical expenses and loss of income [3,11]. Additionally, chronic pain can lead to depression, anxiety, and decreased self-esteem [12,13]. Therefore, effective pain management is crucial for improving the patient’s quality of life following severe trauma [14,15,16]. Multimodal pain management approaches, including pharmacological and non-pharmacological interventions, should be implemented to address the complex nature of pain following trauma, and improve the patient’s overall well-being [17,18].

Persistent enhanced pain after trauma is a common phenomenon that affects a significant proportion of trauma victims. The etiology of chronic pain following trauma is not well understood, but numerous retrospective studies have shown that a significant proportion of chronic pain patients have a history of traumatic injury [19]. Persistent pain after trauma can affect different parts of the body, including the neck ([20]), back [21], shoulder [22], and limbs [19,23]. The risk of developing persistent pain after trauma is higher in females [24]. Psychopathology, such as posttraumatic stress disorder and depression, is associated with persistent enhanced pain in the period immediately following a traumatic injury [25]. Central nervous system changes contribute to the development of persistent pain following surgical trauma and nerve injury [26]. Evidence from a review has indicated that persistent pain is prevalent up to 84 months following traumatic injury [27].

Various terms are employed to describe persistent pain after trauma, and their precise definitions may vary depending on the context. The following are several commonly used terms:Chronic pain: Pain that endures beyond the normal healing period or persists for a minimum of three to six months. It may manifest as continuous or intermittent, and its intensity can range from mild to severe [28].Post-Traumatic Pain Syndrome (PTPS): A condition characterized by sustained pain that emerges subsequent to a traumatic injury or event. This syndrome often encompasses a combination of physical, psychological, and social factors contributing to the perception of pain [29].Complex Regional Pain Syndrome (CRPS): A chronic pain disorder that typically arises after an injury such as a fracture or sprain. It is characterized by enduring severe pain, alterations in skin colour and temperature, swelling, and anomalous hair or nail growth in the affected area [30].Neuropathic pain: Pain resulting from damage or dysfunction of the nervous system. It is frequently described as a shooting, burning, or tingling sensation, and may stem from nerve injuries associated with trauma [31].Central sensitization: A condition in which the central nervous system becomes hypersensitive to pain signals, intensifying the experience of pain. Central sensitization can manifest following trauma and can induce heightened pain responses even in the absence of ongoing tissue damage [32].

However, the current body of research lacks a precise and universally accepted definition for persistent pain after trauma. This study adopts the definition proposed by Macrae and Davies [33,34] in the context of characterizing chronic postsurgical pain. In accordance with this definition, persistent pain is described as pain that meets the following criteria: (1) arises subsequent to a traumatic injury or surgical procedure associated with the injury; (2) endures for a minimum of two months; (3) cannot be attributed to alternative factors such as additional surgical interventions; and (4) is not a continuation of a pre-existing pain condition, which must be ruled out [27].

Regardless of its origin or severity, whether it is long-lasting chronic pain or sudden and intense acute pain, both types of persistent pain can cause considerable physical discomfort and limitations. For example, pain affecting the joints, muscles, or bones can hinder basic activities like walking, standing, or lifting objects, making them challenging to perform [35]. Furthermore, pain can interfere with leisure pursuits such as sports, hobbies, and social interactions, resulting in reduced participation and feelings of isolation. The emotional consequences of pain are equally profound. Individuals dealing with chronic pain often experience anxiety, depression, and a decrease in self-esteem, which can adversely affect their ability to cope with pain and effectively manage their daily lives [36].

Furthermore, understanding the impact of persistent pain is crucial, as it can disrupt various aspects of an individual’s life. Persistent pain can also disrupt sleep, leading to a cycle of fatigue and a lack of energy [37]. This can result in decreased productivity at work, decreased motivation to engage in social activities, and ultimately, further deterioration of physical and mental health. Over time, persistent pain can lead to a decline in overall health and wellbeing. For example, chronic pain can cause physical changes in the brain and nervous system, leading to a higher risk of developing other chronic conditions such as cardiovascular disease, diabetes, and depression. In some cases, persistent pain can even result in disability, making it difficult or impossible to work or perform basic self-care activities.

Clay and colleagues [38] conducted a systematic review to identify early prognostic factors for persistent pain following acute orthopaedic trauma. The review included 23 studies and found that several factors were associated with persistent pain, including pre-existing pain, higher pain intensity at baseline, older age, female gender, and lower education level. The review also found that psychological factors, such as anxiety and depression, were associated with persistent pain. In another very recent review by Alkassabi et al. [39], high pain intensity at baseline, post-traumatic stress syndrome, presence of medical comorbidities, and fear of movement have been identified as significant predictors of persistent pain after trauma. Another recent study [40] found that almost 1 in 2 trauma patients feel daily pain, one year after injury and drug use disorder, alcohol abuse, hospital stay > 5 days, older age, orthopaedic surgery, low education, and extremity injury are significant predictors for persisting pain. Furthermore, patient expectations and coping strategies seem to have a serious impact on persistent pain after trauma as well [41]. Another recent study also underlined the role of mental health factors [42].

In conclusion, identifying prognostic factors early on could help healthcare providers develop targeted interventions to prevent or manage persistent pain following acute orthopaedic trauma. Building upon these considerations, the current study investigates the pain patterns of trauma patients at a German level-one trauma centre, 2 years after trauma. Moreover, the study aimed to identify predictors of persisting enhanced pain, contributing to a comprehensive understanding of the long-term impact of trauma-related pain.

## 2. Materials and Methods

### 2.1. Study Design

The current study is a retrospective single-centre cohort study investigating the pain patterns of patients. Surviving patients were invited to participate in a paper and pencil interview in the second year after trauma. Patients were assessed 23 months (median, IQR = 20–26) after trauma. The aim of the study was to identify predictors for persistent enhanced pain. The study received a positive vote from Witten/Herdecke university’s Ethical Committee (date: 26 April 2018; no. 20/2010). It was conducted in accordance with the Declaration of Helsinki, and all patients were informed that is possible to withdraw their participation at any time.

### 2.2. Questionnaires

#### 2.2.1. Trauma Outcome Profile

The Trauma Outcome Profile (TOP) is a measurement tool for the assessment of the health-related quality of life (HRQoL) of individuals who have sustained serious injuries from trauma. This tool is the trauma-specific part of a larger assessment called the Polytrauma Outcome Chart [43,44] questionnaire. The TOP covers ten dimensions, including depression, anxiety, post-traumatic stress disorder (PTSD), social impact, pain, physical function, daily activities, mental function, body image, and overall satisfaction. Pain and physical function are evaluated using a numerical rating scale (NRS), with scores ranging from 0 to 10, whereby 0 indicates no pain and good function, and 10 indicates worst pain and no function. The NRS scores are recorded for 14 different body regions. If at least one body region was scored >0, it was additionally asked how badly one suffered from pain or functional limitations on a 5-step scale ranging from 0 (not at all) to 5 (extremely). A pain intensity score of 5 or above was considered to require pain therapy. Each of the 10 dimensions of the TOP were transformed into a value ranging from 0 (worst) to 100 (best), where a value of 80 and above corresponds to findings in an average population without serious trauma [43].

#### 2.2.2. AIS—Abbreviated Injury Scale

The Abbreviated Injury Scale (AIS) is a standardized system for describing and classifying injuries based on their severity. It was first developed in the 1960s and has since been revised multiple times to reflect changes in medical knowledge and technology [45]. The AIS assigns a score to each injury based on its severity, ranging from 1 (minor injury) to 6 (a potentially fatal injury). Each injury is classified according to its anatomical location and the type of tissue involved. The AIS is widely used in trauma research and clinical practice to document and compare injury patterns and outcomes across different populations and settings. It is also an important component of the Injury Severity Score (ISS), which is used to assess the overall severity of multiple injuries in trauma patients.

#### 2.2.3. ISS—Injury Severity Score

The Injury Severity Score (ISS) was developed in the 1970s and has since become one of the most widely used scoring systems in trauma evaluation [46]. The scale is based on the Abbreviated Injury Scale (AIS), which is a standardized system for describing and classifying injuries. The ISS is calculated by adding the squares of the three highest AIS scores for different body regions. Each body region is assigned a score between 1 and 6, where 1 represents minor injury and 6 represents a potentially life-threatening injury. The ISS score ranges from 1 to 75, with higher scores indicating more severe injuries. The ISS is particularly useful for triaging trauma patients and determining the appropriate level of care they require. For example, patients with an ISS score of 16 or higher are considered to have severe injuries and are likely to require critical care [47].

### 2.3. Patient Sample

The study sample consists of 635 adult patients (age ≥ 18 years) treated at the Cologne Merheim Medical Centre in the years 2012–2020. The inclusion criteria were adult age >= 18 years and severely injured (ISS 9 + ICU). The exclusion criteria were: death (due to trauma or within 2 years after trauma), patients in a vegetative state (defined by a Glasgow Outcome Scale (GOS) score of 2) or with serious cognitive impairment unable to answer the questionnaire due to trauma sequels or other condition (e.g., severe dementia), lack of German language, denial of participation. Surviving patients were invited to participate in a paper and pencil interview in the second year after admission (follow up rate 50%).

Patients were excluded from data analysis if either the pre-injury or the follow-up pain assessment was missing (*n* = 23). Subsequently, data were checked for plausibility. A considerable increase in pain in body regions that did not fit to the injury pattern were considered unplausible. Sixteen cases were excluded from analysis due to unplausible pain measures. Finally, a total of *n* = 596 patients were included in data analysis.

### 2.4. Statistical Analysis

Prior to data analysis, plausibility checks were performed. If pain scores were missing for individual body regions, we assumed a score of zero. All analyses were performed using SPSS statistical software (version 29, IBM Inc., Armonk, NY, USA). Descriptive statistics are reported as the mean with standard deviation (SD), or as the median with inter-quartile range (IQR), depending on the distribution of the data. Predictors of persisting pain after trauma were investigated by means of logistic regression analysis. The dependent variable for this analysis was relevant pain at follow-up, defined as less than 80 points on the 0–100 pain scale of the TOP. This pain scale is computed using the worst pain score, the sum of pain scores (in 14 body regions), and the level of suffering. In a validation study, 95% of patients with minor trauma reached a value between 80 and 100 points two years after the trauma [43]. The following independent predictors were included in the model: age (3 groups), female sex, relevant pain before the accident, high overall injury severity (ISS ≥ 16), and injured body regions including the head, thorax, abdomen, spine, upper and lower extremities, and pelvis. These body regions were derived from the first digit of the AIS codes, and all injuries with an AIS severity level ≥ 1 were included.

## 3. Results

The mean age of the included patients was 48.2 years (SD 17.8), and 72% (*n* = 428) were males. The mean Injury Severity Score (ISS) was 17.6 (SD 11.8). A small number of cases suffered a penetrating trauma (4%, *n* = 23). The most frequent injury mechanism was traffic accidents (58%), followed by high falls (>3 m height, 17%) and low falls (12%).

### 3.1. Trauma Outcome Profile Pain Scale

Patients had a median pain score of 98.5 (IQR 92–100) prior to trauma, and 8.6% (*n* = 51) scored below the cut-off of 80 points (Figure 1). At follow-up, patients reported a median pain score of 82 (IQR 60–94), and 47.3% scored below the cut-off of 80 points (*n* = 282).

The pain values before and after trauma were clearly correlated (Pearson’s r = 0.40; *p* < 0.001), while the pain score at follow-up only marginally correlated with the ISS (r = −0.10; *p* = 0.011).

### 3.2. Pain Pattern before and after Trauma

Overall, all body areas show a significant worsening in pain severity at follow-up. The most severe aggravation was observed in the body areas of the neck, spine, shoulder, pelvis, hip, knee, and feet (Figure 2).

### 3.3. Injury Severity and Pain

Table 1 shows the mean pain severity score in different body regions, depending on whether this body region was injured (in four subgroups of increasing AIS severity), or not. Also, mild injuries (AIS = 1) show enhanced pain scores.

### 3.4. Predictors of Persistent Pain: Regression Analysis

A logistic regression analysis was performed to determine the predictors of relevant pain after trauma. Relevant pain at follow-up was defined as less than 80 points on the TOP pain scale. Two-hundred and eighty-two patients (47.3%) fulfilled this definition. Table 2 shows the results. With an odds ratio (OR) of 5.5, relevant pain already before the injury was the strongest predictor of persisting pain after trauma. In addition, age was found to be a significant predictor, with those under the age of 30 having the least pain, and those between 30 and 64 years having an OR of 2.4. High overall injury severity, as measured by an Injury Severity Score (ISS) of 16 or higher, was also found to be predictive of persistent pain, with an OR of 1.9. Pelvic trauma was found to be a significant predictor, with an OR of 1.8. Nevertheless, the presence of head and thoracic trauma appeared to contradict the existence of a pain issue during the follow-up period, albeit this observation did not reach statistical significance.

## 4. Discussion

Our study aimed to investigate pain patterns of trauma patients before and after severe trauma, and to identify the predictors of persisting enhanced pain after injury. Our results imply that nearly half of the patients face a reduced pain-associated HrQoL after trauma. Persisting pain seems to be predominant in the body areas of the neck, spine, shoulder, pelvis, hip, knee, and feet. Even minor injuries (>AIS1) lead to increased pain scores after an accident. A major finding of the current study is that pre-existing pain before the injury is a significant predictor of enhanced persisting pain two years after trauma.

### 4.1. Pain before Trauma Is a Strong Predictor for Persistent Enhanced Pain

Our results are in line with earlier studies providing evidence that pre-existing pain is a major predictor of persistent pain after an injury. Patients who suffer from pain before experiencing a traumatic injury are at a higher risk of developing persistent pain following the injury, which can significantly affect their quality of life [27]. Clay et al. [38] identified early prognostic factors for persistent pain following acute orthopaedic trauma in a systematic review of 23 studies. In accordance with our results, one study of this review found “preinjury pain affecting work activities” to be a significant predictor (OR 1.8 (1.3–2.5)) for higher pain severity after trauma [48]. Also, in accordance with earlier research, we were able to identify older age [49,50], female gender [50,51], and high injury severity [52] to be significant predictors of enhanced pain after trauma.

### 4.2. Body Areas of the Neck, Spine, Shoulder, Pelvis, Hip, Knee, and Feet

Regarding body area, our results imply that persistent enhanced pain is especially prevalent in the body areas of the neck, spine, shoulder, pelvis, hip, knee, and feet, which is also in line with earlier research (neck [20], back [21], shoulder [22], and limbs [19,23]). Our study results add the aspect of pelvic injuries, which have been significant predictors of enhanced pain after trauma, while other body regions did not meet the criterion of statistical significance. Other studies reported lower extremity injuries as significant predictors before trauma [53].

### 4.3. Even Minor Injuries (AIS = 1) Lead to Increased Pain Scores after an Accident

In this study, we found that minor injuries cause enhanced pain scores. This is in line with earlier studies reporting that minor injuries, even those rated as AIS1, can lead to increased pain scores after an accident [54,55]. Injuries to the hand and forearm, for example, can generate high costs for society in terms of healthcare and long periods of sick leave [55]. Non-recovery after whiplash was associated with initially reduced cold pressor pain endurance and increased peak pain, suggesting that dysfunction of central pain modulating control systems plays a role in chronic pain after acute whiplash injury [56]. Psychosocial factors, such as posttraumatic stress symptoms, may also raise a major barrier for full recovery of injury patients of even minor levels of severity [57,58].

In conclusion, our research highlights the significance of considering pre-existing pain, the affected body region, and the severity of the injury when evaluating the likelihood of prolonged enhanced pain in patients who have suffered severe trauma. These results have practical implications for enhancing pain treatment and improving the well-being of individuals who have undergone traumatic injuries.

### 4.4. Limitations

Our study is a retrospective, single-centre cohort study. As determined by the design, the results of our study have to be considering in light of the limitation of the design of such studies: single-centre cohort studies have some limitations that should be considered when interpreting their findings. First, the results may not be generalizable to other populations due to the limited sample size and the unique characteristics of the study population. Second, selection bias may be present, as the study participants are often recruited from a single institution or geographic region, and may not represent the broader population. Third, single-centre cohort studies may have limited statistical power to detect small effect sizes or rare outcomes. Fourth, confounding factors may not be adequately controlled for, as the study design may not allow for the adjustment of all potential confounders. Finally, the lack of blinding in the collection of data or the assessment of outcomes may introduce bias into the study results. Therefore, caution should be exercised when interpreting the results of single-centre cohort studies, and their findings should be corroborated by larger and more diverse studies. Additionally, we have to state that the latency between the trauma and questionnaire admission might have caused some bias in the estimation of pain prior to injury.

## 5. Conclusions

Pre-injury pain is a strong predictor of persistent pain in certain body areas after trauma. Effective pain management strategies, including the early identification and treatment of pre-existing pain, are crucial to prevent persistent enhanced pain. Also, mild injury severity may cause persistent enhanced pain and should be considered as an important factor. Further research is needed to identify the most effective interventions to prevent and manage chronic pain in this population.

## Figures and Tables

**Figure 1 medicina-59-01327-f001:**
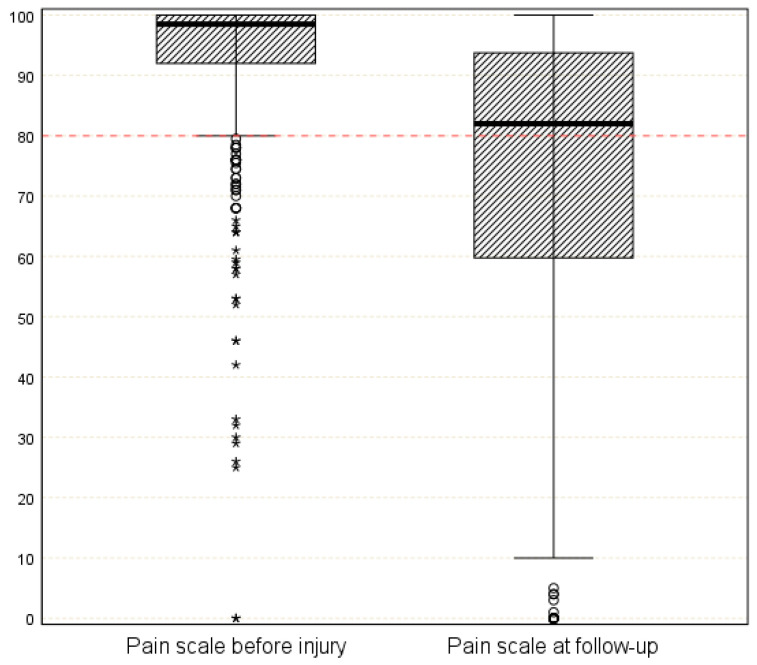
Median pain score as the measured Trauma Outcome Profile before trauma and two years after trauma (o = outliers and * = extreme values).

**Figure 2 medicina-59-01327-f002:**
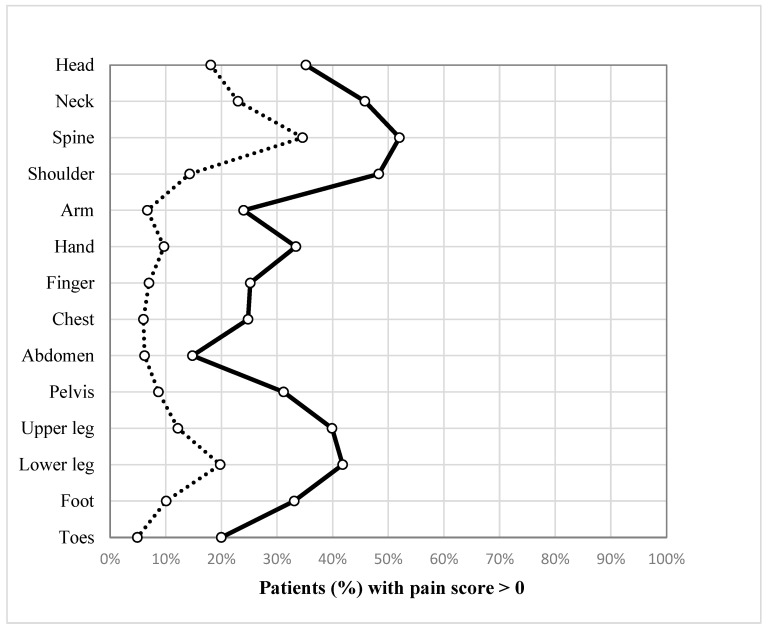
Percentages of patients reporting relevant pain before trauma and two years after trauma for different body regions (dots = percentages; lines = trend pattern).

**Table 1 medicina-59-01327-t001:** Mean pain severity score in different body regions, depending on whether this body region was injured (in four subgroups of increasing AIS severity), or not.

		Injury Severity				
Injured Body Region	Prevalence	Not Injured	AIS 1	AIS 2	AIS 3	AIS 4+
Head	311 (52%)	1.0	1.2	1.5	1.7	2.7
Spinal cord	204 (34%)	1.9	-/-	3.1	3.1	3.6
Thorax	276 (46%)	0.5	0.9	1.0	1.5	1.7
Abdomen	116 (19%)	0.4	-/-	1.0	1.2	1.5
Upper extremity	248 (42%)	2.1	2.6	3.7	4.1	-/-
Pelvis	98 (17%)	1.0	-/-	2.2	3.1	3.6
Lower extremity	226 (38%)	2.2	2.3	4.7	5.0	-/-

-/- less than 10 patients available.

**Table 2 medicina-59-01327-t002:** Results of logistic regression analysis with relevant pain at follow-up (TOP pain scale < 80 points) as the dependent variable (*n* = 596).

Predictor	*n*	Odds Ratio (OR)	95% CI for OR	*p*-Value
Age (reference: <30 years)	122	---	---	0.003
30–64 years	356	2.09	1.33–3.27	0.001
65 and older	118	1.33	0.76–2.34	0.32
Females	168	1.08	0.73–1.59	0.70
Relevant pain before the accident	51	5.43	2.60–11.34	<0.001
ISS 16+	312	1.80	1.20–2.69	0.004
Head injury	334	0.82	0.57–1.18	0.29
Thoracic injury	276	0.70	0.47–1.05	0.082
Injury of the abdomen	116	1.09	0.68–1.76	0.72
Injury of spinal cord	204	1.12	0.77–1.63	0.55
Injury of upper extremity	248	1.22	0.85–1.74	0.28
Injury of lower extremity	226	1.26	0.88–1.82	0.21
Pelvic injury	98	1.96	1.20–3.21	0.008

## Data Availability

Data can be obtained from the corresponding author upon request.

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
