# Peer review of "Pre-Trauma Pain Is the Strongest Predictor of Persistent Enhanced Pain Patterns after Severe Trauma: Results of a Single-Centre Retrospective Study"

_medicina, 2023, doi:10.3390/medicina59071327_

Round 1

Reviewer 1 Report

Thank you for the opportunity to read this manuscript.  Several concerns and questions arise.

How and when was the data concerning preexisting pain collected? When collected two years after the injury when the interview was performed data concerning the status before injury are questionable to rely on their accuracy.

How was the patient cohort chosen? did authors reevaluate all severely injured patients? or just a certain cohort but how did authors determine that cohort? How many did not have increased or prolonged pain? What were the inclusion and exclusion criteria? With a mean ISS of over 16 all patients are supposed to be polytraumatised which would not be surprising for prolonged pain.

Do authors talk about generally pre-existing pain independently of the body region or already preexisting pain in the area of the later injury?

The findings of gender, age and higher injury score are already described. what in new in your study?

The conclusion is primarily aimed at orthopaedic surgeons or doctors who deal with the treatment of chronic pain, but without knowing whether or not those affected will suffer a serious injury in the future... So what are the consequences? And what are the consequences for the treating trauma surgeon who is confronted with acute severe injuries that often require surgery?

what is new and most importantly relevant for the clinically treating trauma surgeon?

Reviewer 2 Report

Abstract
I suggest replacing the Discussion section of the abstract with Conclusions.
1. Include key quantitative findings in the results: The results section of the abstract should provide the reader with key findings from the study e.g. "Older age (odds ratio: x.xx, CI: xx.xx - xx.xx), female gender (odds ratio: x.xx, CI: xx.xx - xx.xx), and high injury severity (odds ratio: x.xx, CI: xx.xx - xx.xx) were identified as significant predictors of enhanced pain."

2. The abstract contains minor grammatical issue. "The most frequent injury mechanism was traffic accidents" should be "The most frequent cause of injury was traffic accidents" or "The most frequent injury mechanism was vehicle accidents"

Introduction

1. Throughout the introduction, the impact of chronic pain on a patient's quality of life is mentioned several times. For instance, lines 45-50 and 78-86 both discuss how pain affects quality of life. Instead, consider summarizing these points once.

2. Line 70-76 - The introduction abruptly jumps into a very specific definition of persistent pain without first providing more context on the various definitions or why this specific one is being used

Methods

Line 110-115 - These are from the template - please carefully check your final manuscript before submission. Remove these lines

Point 2.3 is too vague - Please consider naming this Patient Sample or similar.

1. line 123 - the date mentioned as "26.04.201" appears to be incomplete.

2. line 181 - the reference to Lefering 2012 is not in the standard citation format, which usually includes the title of the paper or work referenced.

Results

Well presented

Discussions

Limitations and strengths of the manuscript are properly compared to the literature.

Conclusion

Concise and well written

References

23 out of 49 references are 10 years old or more. Please consider updating your references.

There are few grammar errors highlighted above.

Round 2

Reviewer 1 Report

Thank you! Some changes have been made, but there are still significant concerns, particularly about the recruitment of the cohort, the accuracy of the retrospective collection of pre-pain data and the clinical conclusion of the manuscript.

Therefore, I leave the decision to the editorial board.

Kind regards

Reviewer 2 Report

Thank you for adjusting the manuscript to a better form as per my recommendations. Article can be published